# Biochemistry of Copper Site Assembly in Heme-Copper Oxidases: A Theme with Variations

**DOI:** 10.3390/ijms20153830

**Published:** 2019-08-05

**Authors:** María-Eugenia Llases, Marcos N. Morgada, Alejandro J. Vila

**Affiliations:** 1Instituto de Biología Molecular y Celular de Rosario (IBR, CONICET-UNR), Ocampo y Esmeralda, Rosario S2000EZP, Argentina; 2Plataforma de Biología Estructural y Metabolómica (PLABEM), Ocampo y Esmeralda, Rosario S2000EZP, Argentina; 3Area Biofísica, Departamento de Química Biológica, Facultad de Ciencias Bioquímicas y Farmacéuticas, Universidad Nacional de Rosario, Suipacha 531, Rosario S2002LRK, Argentina

**Keywords:** copper site assembly, heme-copper oxidases, copper trafficking, metallochaperone

## Abstract

Copper is an essential cofactor for aerobic respiration, since it is required as a redox cofactor in Cytochrome *c* Oxidase (COX). This ancient and highly conserved enzymatic complex from the family of heme-copper oxidase possesses two copper sites: Cu_A_ and Cu_B_. Biosynthesis of the oxidase is a complex, stepwise process that requires a high number of assembly factors. In this review, we summarize the state-of-the-art in the assembly of COX, with special emphasis in the assembly of copper sites. Assembly of the Cu_A_ site is better understood, being at the same time highly variable among organisms. We also discuss the current challenges that prevent the full comprehension of the mechanisms of assembly and the pending issues in the field.

## 1. Introduction

### 1.1. Transition Metal Ions in Nature

First-row transition metal ions are essential for the viability of all living organisms. Organisms from the three domains of life have adopted the utilization of different metal ions as cofactors for challenging enzymatic processes. Taking advantage of their particular chemical properties, transition metal ions are key players in essential metabolic processes like condensation and hydrolysis pathways and (particularly) in redox events based on the different oxidation states displayed by them [1].

These redox properties make them essential but also toxic when their levels are not adequately regulated [2,3]. For example, copper and iron ions are responsible for the formation of toxic hydroxyl radicals due to the Fenton reaction process. Another source of toxicity of copper ions resides in its ability to bind different ligands with high affinity [4]. Thus, when copper uptake is not regulated, copper ions outcompete with native metal ions for their protein binding sites, resulting in mismetallation and disruption of metabolic pathways, ultimately being toxic for the cell. This is the case of Fe-S clusters, among others [5,6,7].

In order to maintain and regulate the homeostasis of transition metal ions, organisms have developed scavenging, storage and metal efflux strategies that are regulated depending on the environmental metal availability [8,9,10]. Bacteria synthesize metal-scavenging siderophores when iron or copper levels are restricted, making metal ions bioavailable or, in the case of pathogenic bacteria, by sequestration of metal ions from the primary immune system response [11,12,13,14]. Bacteria have also metal storage proteins like ferritin or metallothioneins. These proteins are able to bind strongly iron and copper ions and function as detoxifying proteins, precluding the formation of reactive oxygen species (ROS) [15,16]. When the cytosolic levels of metal ions raise beyond certain levels, bacteria synthesize metal efflux pumps that favor the release of free metals ions that can cause DNA damage [17].

Eukaryotes have evolved more sophisticated regulatory pathways. From a cellular point of view, metal homeostasis depends on mechanisms similar to those present in bacteria. Metal uptake, storage and efflux are tightly regulated [8,18]. Cells regulate the localization and maintenance of copper pools to ensure the activity of copper-dependent proteins, avoiding toxic accumulation of copper in the nucleus or in the cytoplasm [19]. In the case of insufficient metal levels, the metal ions contained in the native cellular metal reservoir are targeted to the organelles that require them to guarantee the survival of the cell. In the case of copper, two main destinies have been identified: the mitochondria, where cellular respiration takes place, and the chloroplasts in the case of plants [20,21,22]. When facing an excess of cytoplasmic metal levels, storage proteins like metallothioneins and ferritin work in a similar manner to bacteria binding the metal ions. An excess of metal ions can also be pumped outside the cell by means of specific metal transporters [23].

Additionally, superior eukaryotes ensure the systemic metal homeostasis by several regulation mechanisms. First, the incorporation of dietary iron and copper has to be regulated in order to limit the uptake of these metals to trace amounts and avoid toxicity. Second, the metal ions are transported through the blood system by specific metal carriers to ensure the delivery of metals to the cells. And last, the metal ion incorporation is regulated by the expression of carrier interacting proteins in the surface of the cells, to mediate metal ion delivery to the required cells [10].

### 1.2. Copper is Essential for Terminal Oxidases

Copper is essential as a metal cofactor in different essential cycles. Copper can cycle between two oxidation states in the cell: Cu^+1^/Cu^+2^. This one-electron reaction is used in nature in different catalytic processes that span a wide range of redox potentials (200 to 1000 mV) based on the influence of the metal ligands and the protein environment that modulate the electronic structure of the copper site and, consequently, its reactivity features. For example, in the oxygen cycle, copper is involved both in the reduction of oxygen (aerobic respiration), but also in the formation of molecular oxygen in the photosynthetic pathway [22,24]. Copper is also involved in the nitrogen cycle as the main cofactor in the terminal anaerobic oxidase N_2_O reductase [25].

Cytochrome *c* oxidases (COX’s) are protein complexes from the superfamily of heme-copper oxidases responsible for most of the molecular oxygen reduction in nature. Members of this superfamily also include the bacterial quinol oxidases, but terminal oxidases accepting electrons from cytochrome *c* are more spread from bacteria to superior eukaryotes.

COX’s are integral membrane proteins complexes that show different degrees of complexity depending on the organism. Bacterial oxidases are embedded in the plasma membrane and contain 3–4 subunits, while eukaryotic oxidases are located in the inner membrane of the mitochondria, where cellular respiration takes place [26,27]. Eukaryotic enzymes are more complex than the bacterial ones, being able to harbor as much as 13 subunits, as observed for the bovine and human oxidases. However, the catalytic core of the oxidases consists only of subunits I to III, highly conserved in all organisms. The *ba_3_* oxidase from *Thermus thermophilus* is one of the simplest known oxidases, containing only these three subunits (Figure 1a) [26].

Copper ions fulfill two essential roles in COXs, distributed into two sites. The Cu_A_ site, located in subunit II (COX II) acts as the electron acceptor from cytochrome *c*. The Cu_B_ site, located in subunit I (COX I) is the oxygen binding site that catalyzes its reduction to two water molecules, completing the cellular respiration process. Both copper centers are essential for oxidase activity, and therefore for cellular respiration [28,29].

The Cu_B_ site is tightly associated to an iron from a *a_3_* heme [30], except in the case of the *bb_3_* oxidase of *T. thermophilus* and the *ba_3_* quinol oxidase of *E. coli* (Figure 1c). This copper ion is coordinated by three conserved His ligands, with the particularity that one of them is covalently bound to a proximal Tyr residue. This center is involved in one-electron transfer processes since it fluctuates between the Cu_B_^+2^ and Cu_B_^+1^ oxidation states [22,24].

The Cu_A_ site is a binuclear copper site (Figure 1d). Two copper ions are bridged by two thiols moieties from Cys residues. The coordination sphere is completed by one His residue bounded to each copper ion, and weak axial ligands that impose an asymmetry to the metal site. One copper is coordinated by an oxygen from a backbone carbonyl moiety while the other copper ion is coordinated to the thioether moiety of a Met residue [22,24]. This coordination is fully conserved in cytochrome *c* oxidases, and also in Cu_A_ centers from N_2_O reductases [31]. Despite having two copper ions, the Cu_A_ site is involved in one-electron transfer processes. The accessible oxidation states are the reduced Cu_A_^+1^ (with two reduced Cu^+1^ ions) and the oxidized Cu_A_^+3^ (forming a mixed-valence copper site where each Cu ion has a +1.5 formal charge) [25].

### 1.3. Assembly of COX Catalytic Core

Yeast was used as the initial model organism for the study of the assembly of the oxidase. Knock out experiments allowed the discovery of essential proteins required for the assembly of a functional COX and the identification of different subcomplex maturation states of COX. These intermediate subcomplexes allow the definition of critical steps in the assembly process, with the first two ones involving the appropriate formation of the metal centers in the catalytic core required for the activity of the protein [32].

The first stage involves the complete maturation of COX I, an integral membrane protein that spans 12 times across the membrane. COX I contains two heme moieties and the Cu_B_ site (Figure 1c). These heme moieties can vary among different organisms and are optimized for the oxygen levels of their environment [33]. The second stage requires the correct maturation of COX II. COX II is anchored by one or two transmembrane helixes, with the Cu_A_ site being placed in the soluble domain, where the interaction with the soluble cytochrome *c* takes place (Figure 1d) [34]. Once both copper cofactors are correctly assembled, the maturation of COX continues with the binding of COX III, an integral membrane protein that stabilizes the protein complex and is devoid of metal cofactors (Figure 1a–b) [32].

In eukaryotes, the expression of COX is a highly coordinated and regulated process that requires the expression of genes from different origins. Genes encoding the catalytic core of the oxidase (COX I-III) are within the mitochondrial DNA, while the remaining subunits (IV to VIII) are encoded in the nuclear DNA [35,36]. Additional gene products are required to control the correct assembly of the complex and their cofactors with different origins. Indeed, up to 30 assembly factors have been identified in mitochondria [37,38].

A homology search has led the definition of possible assembly factors in bacteria [39,40]. The genes coding for the three essential subunits and the accessory proteins required for cofactor assembly are clustered within the same operon. The large variability of bacterial species has also led to the adaptation of different sets of proteins to fulfill all the functions required for the assembly of a functional oxidase.

In this article, we review the information available about the biochemical process of assembly of the two copper sites in heme-copper oxidases in different organisms, highlighting the differences and similarities between bacterial and mitochondrial systems.

## 2. Maturation of COX I and the Cu_B_ Site

### 2.1. Translation and Membrane Insertion of Apo-COX I

The correct maturation of this protein is the first checkpoint in the assembly of the oxidase. Eukaryotic and bacterial COX I synthesis and membrane insertion depend on insertion machinery of the inner mitochondrial membrane or periplasmic membrane, with the assistance of additional membrane proteins in eukaryotes that stabilize the recently synthesized COX I, Cox14 and Coa3. These proteins, not present in bacteria, have been identified to interact with newly synthesized COX I to prevent degradation and allow the correct assembly of the metal cofactors and posterior binding of additional COX subunits [41,42].

### 2.2. Heme Insertion into COX I

The correct insertion of the heme moieties is essential for the functionality of COX. In mitochondria and bacteria, failures in the heme synthesis or in the heme insertion factors lead to accumulation of the apo protein [43,44]. As mentioned before, iron is a toxic ion, and in order to avoid the occurrence of toxic reactions the heme moieties are not free. The synthesis of the heme moieties and COX assembly takes place within the same subcellular localization. From bacteria to eukaryotes, heme *a* is of exclusive use of the terminal oxidases complexes and its synthesis take place within the same subcellular localization as COX assembly [45]. Its formation is catalyzed by Cox 10 and Cox15. These proteins are integral membrane proteins located in the cytoplasmic membrane of bacteria or the inner mitochondrial membrane (IMM) of eukaryotes[46,47,48,49]. Then, the heme moiety is transferred to COX I by SurfI, a highly conserved protein from bacteria to superior eukaryotes. SurfI is a membrane anchored protein with two transmembrane helices, one on its N-terminus and the other at the C-terminus with the soluble domain facing the periplasm or the inter-membrane space (IMS) [50]. This protein transfers the heme moieties to their COX I binding sites while avoiding the release of the free cofactor [51,52,53].

Some questions are still unanswered about this process. Since the heme moieties are placed deep inside the transmembrane helices of COX I (Figure 1), the heme transfer is a challenging process. Despite the accumulation of apo COX I, the cotranslational insertion of the heme moieties cannot be discarded. The complexity of the system makes the biochemical evaluation of this process still unavailable.

### 2.3. Cu Insertion to COX I

The metallochaperone for the Cu_B_ site in most organisms is from the Cox11/CtaG protein family [54,55,56]. Deletion of this protein leads to absence of the Cu_B_ site in most organisms and accumulation of apo COX I [29]. Cox11 is anchored to the membrane by its N-terminal domain, with a soluble C-terminal domain facing the periplasm in bacteria or the IMS in eukaryotes. This protein functions as a dimer, with two copper ions in a Cu_2_S_4_ metal cluster in the protein-protein interphase with each monomer contributing with two sulfur moieties from Cys residues in a CysXCys conserved domain (Figure 2) [57,58,59]. The copper transfer mechanism is not well understood, but it is proposed Cox11 transfers one copper ion to COX I post-translationally.

Until now, there are not clear biochemical experiments demonstrating the metallochaperone function of Cox11 because of the complexity of the system. The main challenge to pursue this aim is the difficulty in expressing and handling subunit COX I. However, the study of the interaction between Cox11 and COX I is a relevant pending issue. Several questions about this process are still unanswered: (1) Which is the copper ion being transferred? (2) Why does this protein bind two copper ions to perform a one-copper ion transfer? (3) How is the copper inserted in its native site buried within the protein structure?

### 2.4. Sco1 as a Copper Donor

In some bacteria lacking a Cu_A_–depending oxidases, like *R. capsulatus* and *P. aeruginosa*, the functionality of their oxidases is affected by mutation in proteins from the Sco family. These proteins are mainly involved in the assembly of the Cu_A_ site (vide infra). However, mutation of Sco proteins in these bacteria impairs the oxidase activity. In all cases, the native phenotype can be recovered by the addition of copper to the growing milieu. These results suggest that Sco proteins can also be involved in the assembly of the Cu_B_ site, but with a different mechanism compared to that involving Cox11 [60,61].

### 2.5. Heme Moieties and Protein Functionality

Bacterial and eukaryotic oxidases differ by the number of subunits, the amount of accessory proteins involved in the stabilization during the maturation of COX I, and also by the chemical nature of the heme moieties. For example, *T. thermophilus* possesses two distinct terminal oxidases: *caa_3_* COX and *ba_3_* COX. The main structural difference between these proteins is that the donor cytochrome *c* is fused to subunit II in the *caa_3_* oxidase. The presence of this fusion does not abide electron transfer from the soluble cytochrome *c_554_*, the common electron donor in bacterial oxidases. Despite both oxidases share the same catalytic center, the *ba_3_* complex is almost exclusively used under low oxygen atmosphere while the *caa_3_* complex is the main oxidase under normal oxygen conditions.

This heme variation results in an increased oxygen affinity by the oxidase. This variation also allows bacteria to use different final electron acceptors like nitric oxide under reducing conditions. This can be achieved by the *T. thermophilus ba_3_* oxidase or by changes in the active site by insertion of a *b_3_* heme moiety to form a binuclear site with Cu_B_ like the *bb_3_* oxidase form *P. stutzeri*. Modification of the heme moieties is common in the COX-related quinol oxidases that can change from a *bo_3_*-Cu_B_ oxidase under normal oxygen conditions to *bd_3_* quinol oxidases without the requirement of the Cu_B_ site under low oxygen pressures.

## 3. Maturation of COX II and the Cu_A_ Site

### 3.1. Translation and Membrane Insertion of Apo-COX II

COX II is cotranslationally translocated and inserted into the membrane by the highly conserved family of OXA transporters. The Oxa family is an ancient group of transporters of bacterial origin that are conserved in mitochondrial genomes and whose function is the insertion and translocation of proteins to their native environment [62]. In bacteria, Oxa homologs called YidC are responsible for the insertion of the recently synthesized polypeptide in the membrane and for the translocation of the soluble domain [63]. In fungi and animals, two proteins from the Oxa family have been described: Oxa1 and CoxOxa1 inserts the nascent N-terminal transmembrane helix, and Cox18, which is specific for Cox II, translocates the C-terminal soluble domain [62]. Plants contain four Oxa homologs, among them, Oxa1a is the N-terminal insertase and Oxa2b is the translocase [64].

In addition to these two transporters, a number of chaperones that participate in the insertion and folding of COX II have been reported. In yeast and mammals, the newly synthesized COX II initially interacts with chaperones Mba1 and Cox20 [65]. None of these chaperones has been reported in plants or in prokaryotes. Their function is to stabilize the newly synthesized apo-protein and recruit the group of accessory proteins responsible for the assembly of the Cu_A_ site, collectively known as the “metallochaperone module” [34].

### 3.2. The Metallochaperone Module and the Assembly of the Cu_A_ Site

The proteins from the “metallochaperone module” act in coordination, and this fact has thwarted the functional annotation of each one separately solely based on knockout experiments. Despite the protein composition of this module is variable between species; it is always composed by proteins belonging to a few families. Even when the proteins involved in this module are structurally related in different species, they can display a high functional diversity and also overlapped roles that make it difficult to determine of the activity of each one. The proteins forming the metallochaperone modules reported in several species are summarized in Figure 3 and Table 1.

This module performs two functions: first, to ensure that the coordinating cysteines from Cox II are reduced and able to bind copper, and second, to insert the copper ions into Cox II to generate a functional Cu_A_ site (Figure 3a). The mechanisms by which these functions are achieved are still under debate.

### 3.3. Cu_A_ Assembly in Bacteria

Gram-positive bacteria contain heme-copper oxidases with Cu_A_ as the electron entry port, located in the outer face of the plasma membrane. The assembly of this copper site has been extensively studied in *B. subtilis*; that was the first bacteria in which Cu_A_ assembly proteins were identified. In this organism, Cu_A_ maturation requires one Sco ortholog (YpmQ) [39]. Sco (from synthesis of cytochrome oxidase) is the name of an ancient and highly conserved family of proteins with a soluble domain adopting a thioredoxin fold attached to the plasma membrane (or the inner mitochondrial membrane in eukaryotes) by a single helix, and is able to bind copper through a CysXXXCysX_n_His motif [66] (Figure 4). Sco proteins differ from canonical thioredoxins by having an additional loop (known as the “Sco loop”) that contains the extra His in this motif, that endows these proteins with the metal binding ability that thioredoxins normally lack. In copper-bound Sco proteins, this loop adopts a β-hairpin conformation. This does not preclude these proteins to act as thiol-disulfide oxidoreductase, since the thiol groups of the Cys residues can be oxidized to form a disulfide bond. In principle, based on these features, Sco proteins would be able to perform sulfur-based redox 0chemistry and act as metallochaperones.

Knockout of Sco on *B. subtilis* gave rise to deficient strains unable to assemble the Cu_A_ site of COX. This deficiency can be rescued by adding excess copper to the medium, suggesting that this protein could act as a copper metallochaperone (Figure 3b) [39]. *B. subtilis* Sco is able to bind both Cu^+1^ and Cu^+2^ in vitro, and there is an ongoing discussion suggesting that Sco proteins could transfer either Cu^+1^ and Cu^+2^. Since its discovery, extensive structural and biochemical studies have been performed, and different possible mechanisms of copper insertion have been proposed [67,68,69,70,71]. However, the copper insertion on Cu_A_ site has not been proven at the biochemical level.

Another widely studied gram-positive bacteria is *Streptomyces lividans*, whose metallochaperone module is composed by Sco and the extracytoplasmic protein ECuC, which is an ortholog of pCuAC (periplasmic Cu_A_ chaperone) (Figure 3c) [72,73]. The members of pCuAC family are copper-binding proteins present in many bacteria. They present a cupredoxin-like fold with the insertion of a solvent-exposed β-hairpin. These proteins contain a Cu(I) binding site involving two methionines and two histidines in a highly conserved H(M)X_10_MX_21_HXM motif that is defined between the cupredoxin domain and the β-hairpin (Figure 4) [73,74]. These proteins are generally assumed to be copper metallochaperones based on structural similarity with *T. thermophilus* pCuAC (see below). This strongly supports the annotation of these proteins as copper chaperones, despite their essentiality seems to be strongly organism-dependent.

By in vivo and in vitro studies, Blundell et al. have demonstrated that *S. lividans* Sco is devoid of thiol-reductase activity, but is able to act as the copper chaperone for the Cu_A_ site, being critical for copper capture from the extracellular medium and subsequent transfer to Cox II. However, at the moment, it is not clear yet whether Sco simply sequesters the copper ions from the medium or ECuC is a copper donor for Sco, acting upstream in the mechanism of copper acquisition. *S. lividans* Sco is critical for morphological development and copper homeostasis, suggesting that this protein also plays regulatory roles [72].

The mechanism of Cu_A_ assembly of the *ba_3_* oxidase from *Thermus thermophilus* was the first one described at the biochemical level, in which two periplasmic proteins were identified as assembly factors for this site. These proteins are orthologs of the proteins identified for *S. lividans*: a periplasmic soluble copper binding protein pCuAC (related to ECuC) [77], and a protein from the Sco family. In vitro experiments performed with purified proteins have shown that *T. thermophilus* Sco is unable to deliver the copper ions to the soluble domain of COX II, but instead it reduces the cysteine residues. Afterwards, pCuAC acts as the periplasmic copper chaperone inserting sequentially the two required copper ions (Figure 3d) [74]. These biochemical experiments were performed with purified, soluble proteins. This is feasible due to the high stability of the soluble domain of *Thermus thermophilus* COX II, that contains only the Cu_A_ cofactor and there are no interferences from the Cu_B_ site and the heme groups.

Among gram-negative microorganisms, alpha-proteobacteria are particularly relevant for the study of Cox assembly factors based on their evolutionary relationship with eukaryotic mitochondria and provide a simple working model. Next, we will review the state-of-the-art and the perspectives regarding Cu_A_ assembly in two important model organisms from this group: *Paracoccus denitrificans* and *Bradyrhizobium japonicum*.

*Paracoccus denitrificans* expresses two Sco orthologs (ScoA and ScoB), and two periplasmic copper chaperones related to pCuAC (pCu1 and pCu2) [78]. Deletion of these proteins individually or by pairs does not have any effect in *P. denitrificans* COX activity under normal copper levels. However, when bacteria are grown on low-copper media, ScoB becomes critical for Cu_A_ assembly. This suggests that ScoB is the copper chaperone for Cu_A_ site (Figure 3e). Despite ScoA have no observable phenotype by its own, the double mutant ΔΔScoA/B displays diminished COX activity and impaired Cu_A_ metallation. Based on this evidence, authors propose that ScoA might have some role assisting ScoB on copper delivery. One of the possibilities would be ScoA being a thiol-reductase for ScoB and/or CoxII. No specific role has been assigned to any pCu protein via knock-out experiments, suggesting that these proteins are not essential, even under copper starvation conditions [78,79]. Further research is necessary to address whether they act in the assembly of copper sites or in other copper homeostasis processes.

The case of *Bradyrhizobium japonicum* is more complex. This gram-negative bacteria encodes two periplasmic pCuAC-like proteins, pCuA and pCuC, located in a copper-inducible operon that also contains a membrane copper receptor and a cytoplasmic transporter [80]. Also, *B. japonicum* expresses a membrane-anchored thioredoxin-like protein, TlpA, and a Sco ortholog, ScoI. TlpA was originally annotated as a thiol-reductase since its discovery [81,82] but only recently their substrates have been identified: TlpA reduces both ScoI and COX II [83]. The mechanisms of this reduction have been described with exquisite structural detail by Abicht et al., who managed to crystallize the mixed disulfide intermediates of TlpA with both targets, and described the interaction surfaces [84]. The copper metallochaperone for the Cu_A_ site in *B. japonicum* remains to be elucidated, the obvious candidates being ScoI and the two periplasmic chaperones pCuA and pCuC (Figure 3f). ScoI has been shown to bind Cu(II) but the transfer to COX II could not be observed [85]. Further research is required to identify the copper chaperone for COX II in this system, and particularly further in vitro studies, since the phenotypes of all the deletion mutants are very similar.

### 3.4. Cu_A_ Assembly in Mitochondria

Most of the assembly factors of the Cu_A_ site described for bacteria are conserved in mitochondria. However, the mitochondrial metallochaperone modules are much more complex and diverse among different organisms. The roles of the different factors are even more variable, resulting in different mechanisms. Here we will summarize the information available in eukaryotes, which involves yeast, human and plant mitochondria.

Two common features in the mitochondrial Cu_A_ assembly is that they usually have more than one Sco protein and that they lack a homologue of the periplasmic copper chaperones from the pCuAC family. Instead, a soluble copper chaperone named Cox17 conserved in the IMS in yeast [86], human [87] and plants fulfills [88] this function. Cox17 is a small, cysteine-rich soluble protein essential for mitochondrial copper trafficking in all studied eukaryotes. Structurally, it presents two alpha helixes and a long N-terminal unstructured region that binds Cu(I) by a CX_3_X motif (Figure 4c) [89]. Cox17 has been shown to transfer copper to both human Sco1 and human Sco2 but using different mechanisms. Cox17 is able to transfer one equivalent of Cu(I) to Sco1 in an event coupled with a redox reaction in which the disulfide bond in apo, oxidized Sco1 is reduced, thus promoting metal transfer. Instead, Cox17 directly transfers Cu(I) to reduced Sco2 (Figure 3h) [86,90]. Additionally, Cox17 has been related with a regulatory and structural function in mitochondria, playing a role in the biogenesis of the cristae organizing system [91].

The metallochaperone module in yeast is formed by two Sco proteins: Sco1 and Sco2 [92]. Only Sco1 has been shown to be essential for COX biogenesis, and it has been suggested to be the metallochaperone [93,94], despite biochemical studies are not available due to the difficulty in obtaining a soluble yeast COX II domain. Additionally, Coa6 and Cox12 have been identified as essential components of the Cu_A_ assembly machinery, with probably overlapping roles (Figure 3g) [95].

Human mitochondria expresses also two Sco orthologs, both of them being essential for proper metallation of the Cu_A_ site [28] and to elicit a functional oxidase. Mutants in Sco1 and Sco2 have been shown to give rise to lethal pathologies that are tissue specific, in all cases leading to a non-functional oxidase [96,97,98]. Genetic studies have shown that Sco2 acts upstream to Sco1 playing complementary roles. Biochemical experiments later demonstrated that Sco2 is a copper-dependent thiol-reductase, and Sco1 is a copper metallochaperone [99]. Another key factor of the metallochaperone module is Coa6, a soluble protein which is essential in some cell types [100,101], and it is proposed to have overlapping roles with Sco2 [95]. Human cells also express an ortholog of yeast Cox12, named Cox6b, that appears to be involved also in the metallochaperone module, but its role and importance are still uncertain [95]. Cox16 is a small transmembrane protein and although it does not have a copper binding motif, it is involved in the metallochaperone module, being required for the interaction of Sco1 with Cox II and also for the assembly of Cox I and Cox II [102,103]. Surprisingly, Cox16 is also present in yeast, but appears to have a different role, not associated with Cox II biogenesis [104,105,106]. No homologs of Cox16 are found in *A. thaliana*.

Plants present a different scenario compared to the yeast and human metallochaperone modules. They express two homologs of Cox17, that apparently have redundant functions and are not essential for COX activity, but influence stress responses [88]. Plants also express two Sco orthologs, Hcc1 and HccHcc1 is essential for Cox II activity and is the only one with the canonical copper-binding motif. Biochemical experiments have shown that Hcc1 is a Sco protein that is able to act as the metallochaperone for plant CoxII (Figure 3i) [75]. In contrast, Hcc2 does not contain the typical copper binding motif nor the redox active cysteines present in Sco proteins. Knock out experiments in *A. thaliana* have shown that Hcc2 is not essential for the oxidase activity. However, Hcc2 is expressed in plant mitochondria and interacts with the metallochaperone module, probably assisting HccAn additional role related with regulation of redox homeostasis has also been proposed for HccA recent work shows that surprisingly Hcc2 is able to complement a Sco2 deficient yeast mutant, despite its lacks the copper binding motif [107]. More work is required to identify the differences in plant mitochondria compared to other higher eukaryotes.

### 3.5. A Strategy for the Biochemical Analysis of the Mitochondrial Metallochaperone Module

Many progresses in the functional annotation of each implicated protein have been achieved by elegant in-vivo experiments, but the precise determination of their function and mechanisms must come invariably from in-vitro biophysical and biochemical data. One of the biggest obstacles to overcome is that eukaryotic COX II subunits are unstable, making them difficult to purify and perform biochemical studies. At the moment, there are no reports of a purified, stable soluble domain of a eukaryotic COX II. This difficulty can be rationalized by analyzing the structures of the oxidases and the respiratory complexes (Figure 1a,b). The functional domain of COX II containing the Cu_A_ site presents a typical cupredoxin fold, and it is located outside of the membrane. The bacterial COX complex only contains the three catalytic core subunits (Figure 1) in such a way that the globular domain is mostly exposed to the solvent. Twenty years ago, the structure of bovine COX was determined by X-ray crystallography (Figure 1) [27]. Recently, cryo-electron microscopy allowed the structure determination for the respiratory supercomplexes of both yeast and human mitochondria [108,109]. All these structures show that the globular domain of COX II in eukaryotic COX is buried within the complex. The overall folds of the mitochondrially encoded catalytic subunits are virtually identical to the bacterial ones, but the mitochondrial complexes contain several accessory subunits (8 for yeast, 11 for human) with structural functions around the catalytic core. As a result, the COX II surface is much less solvent exposed, and therefore the mitochondrial globular domains are expected to be less stable in solution.

A strategy for the biochemical study of eukaryotic COX II resorted to protein engineering, by designing a chimeric protein using the soluble domain of *T. thermophilus* COX II as a scaffold. The native loops from this bacterial protein were replaced by the loops from the human and plant COX II oxidases. These chimeric proteins were expressed and purified and displayed the exceptional stability of bacterial COX II, and a native-like Cu_A_ site [110]. These chimeric soluble domains were exploited to follow the thiol reduction and copper transfer steps with different proteins from the metallochaperone module. These experiments allowed the functional annotation of human Sco1 and Sco2 [99], and plant Hcc1 [75].

Copper transfer in vitro was sensitive to the sequence of the replaced loops. Human Sco1 was able to transfer copper to the chimeric protein with the human-like Cox II loops, but the reaction did not proceed toward the bacterial Cox II domain [99]. These results demonstrate that loop recognition is key for the functions of thiol reduction and copper insertion by Sco proteins. This role of the loops was early suggested by NMR experiments that showed that, in the absence of metal ions, are in dynamic equilibrium, providing a recognition site for the metallochaperone [111].

Further insights about the importance of the copper binding loops on chaperone recognition provide from in-silico structural models such as the simulation of the interaction between Cox20 and COX II [112]. In this structural model, Cox 20 associates with COX II by the zone immediately adjacent to the copper binding loops. This model, in addition with the loop specificity observed on purified proteins, fits well with the function of Cox 20 as a recruiter of the metallochaperone module.

## 4. Concluding Remarks and Perspectives

The function of COX is one of the most important processes for the cellular cycle in aerobic organisms. The different mechanisms that ensure the correct assembly of this enzyme have been extensively studied over the past years; however, there are still severalimportant issues that remain to be elucidated.

Soluble metallochaperones acting upstream of Cox11 and the Cu_A_ metallochaperone module have shown to be conserved, and essential in most cases. These chaperones include the pCuAC family among bacteria and Cox17 in eukaryotes. A pending issue is to determine the mechanisms of copper transfer and the target proteins for copper insertion of each ortholog.

The assembly machinery of the Cu_B_ site is highly conserved among all studied organisms, although the biochemistry of the mechanism is not well understood. The main difficulty for studying the mechanisms on this system is that all the involved factors are membrane proteins. The availability of a stable model of apo COX I would be highly relevant to undermine the mechanistic insights of the formation of the different metal cofactors in this protein.

Assembly of the binuclear Cu_A_ center has been extensively analyzed in different organisms since its first description in the 80’s. The biosynthesis of this site has shown to be surprisingly complex, involving a still growing number of involved assembly factors.

The Sco family includes the most important players in the assembly of the Cu_A_ site, and the only one being essential in almost all analyzed organisms. The diversity in the biochemical properties and function of Sco proteins from different organisms is an intriguing topic. One or two orthologs of Sco have been shown to be required, and in different organisms they perform different functions, even when are highly similar in structure. The function of this family has been often related with the cysteine residues and the ability of copper binding. The recent report of Sco family members as Hcc2 (or *At* Sco2) that lack these structural features and still can complement the lack of other Sco proteins is very surprising. This highlights that these proteins are also performing a different role, apart from the typical thioredoxin/metallochaperone activities. Further research is required to assess which is this function.

One of the most significant challenges is to identify the biochemistry of each protein separately, and to describe how they cooperate to fulfill their function. In this review, we have discussed the high diversity and complexity of the mechanisms of assembly of COX copper sites, even when the COX structural subunits are highly conserved. This diversity encompasses changes in the required assembly factors upon exogenous conditions, or even in different tissues of the same organism. This picture discloses how cells have evolved to provide alternative routes to guarantee the assembly of essential copper sites under different (sometimes challenging) conditions.

## Figures and Tables

**Figure 1 ijms-20-03830-f001:**
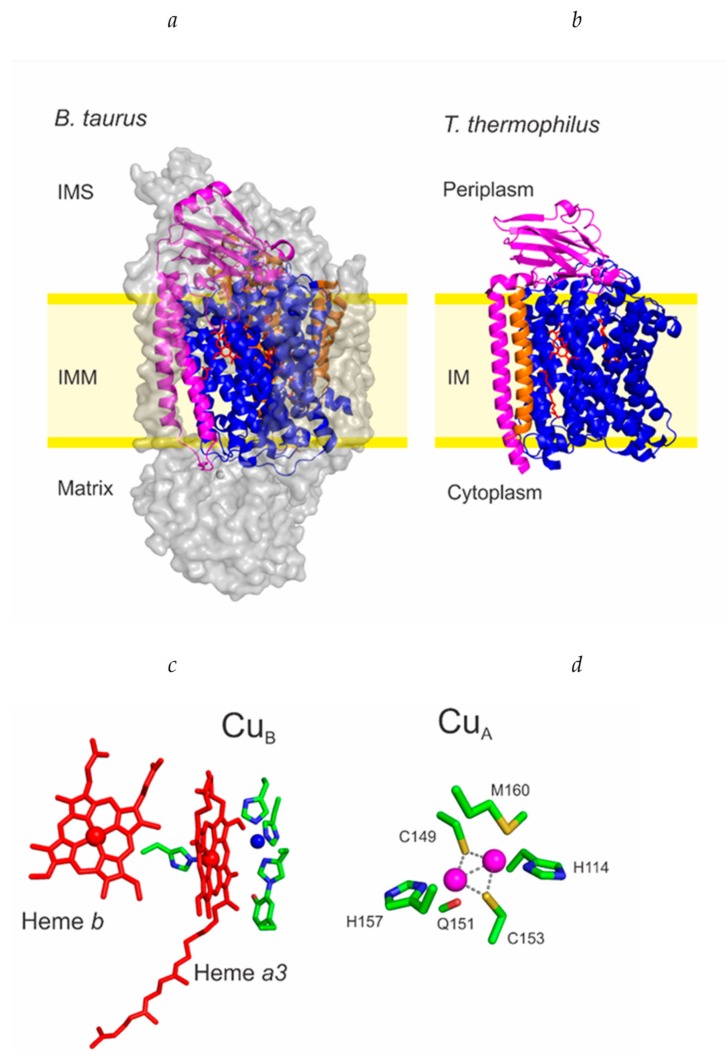
Structures of the oxidases and the metal binding sites: (**a**) Crystal structures of the bovine cytochrome *c* oxidase (PDB ID 5B1A); (**b**) Crystal structure of the *ba_3_* oxidase from *Thermus thermophilus* (PDB ID 3S8F). The different colors highlight the three essential subunits: COX I (blue), COX II (purple) and COX III (Orange). (**c**) Structure of the metal cofactors in COX I: Heme *b*, heme *a_3_* and Cu_B_ site (**d**); Structure of the Cu_A_ site in COX II. Residue numbering corresponds to the *ba_3_* oxidase from *Thermus thermophilus*.

**Figure 2 ijms-20-03830-f002:**
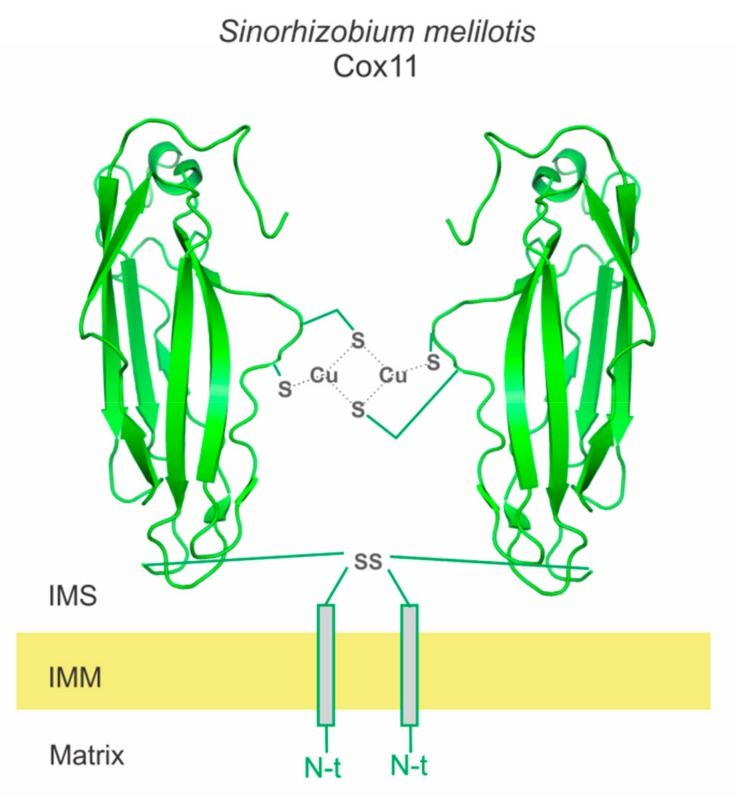
The Cu-Cox11 dimeric form. Ribbons depicts the solution NMR structure of Cox11 (PDB ID 1SP0) from *Sinorhizobium melilotis* [57].

**Figure 3 ijms-20-03830-f003:**
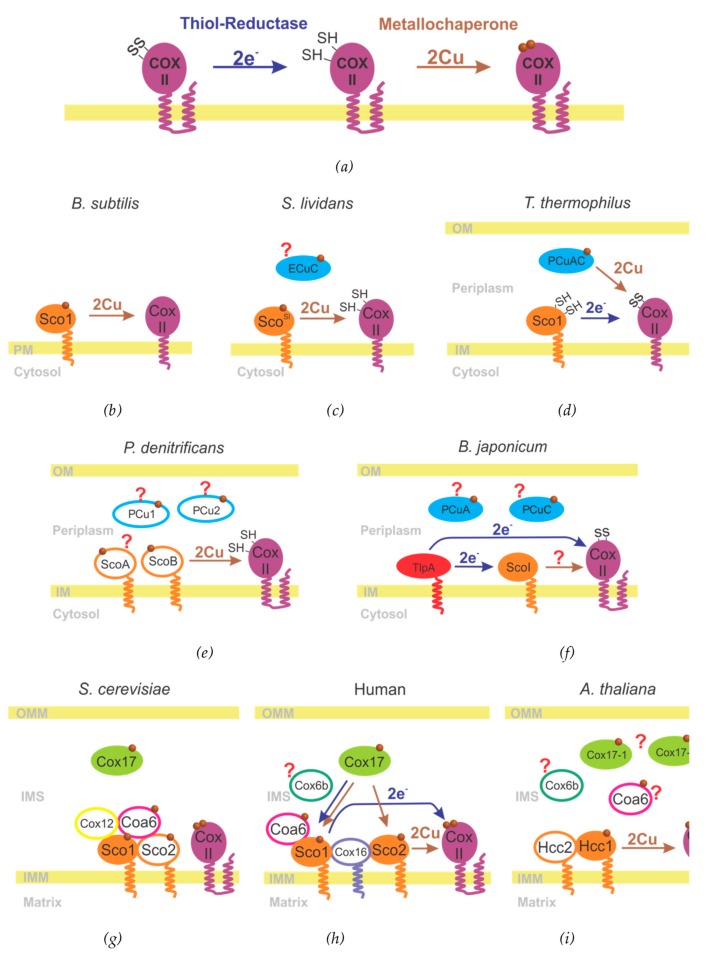
Mechanism of Cu_A_ assembly in different organisms. (**a**) The activities required to assemble the Cu_A_ site on newly synthesized COX II. (**b**–**i**) Metallochaperone modules in different organisms are indicated from. Proteins from the same families are depicted in the same color. Essential proteins have been colored in filled ovals, and non-essential ones as empty ovals. Arrows indicate the function of each proteins, when assigned.

**Figure 4 ijms-20-03830-f004:**
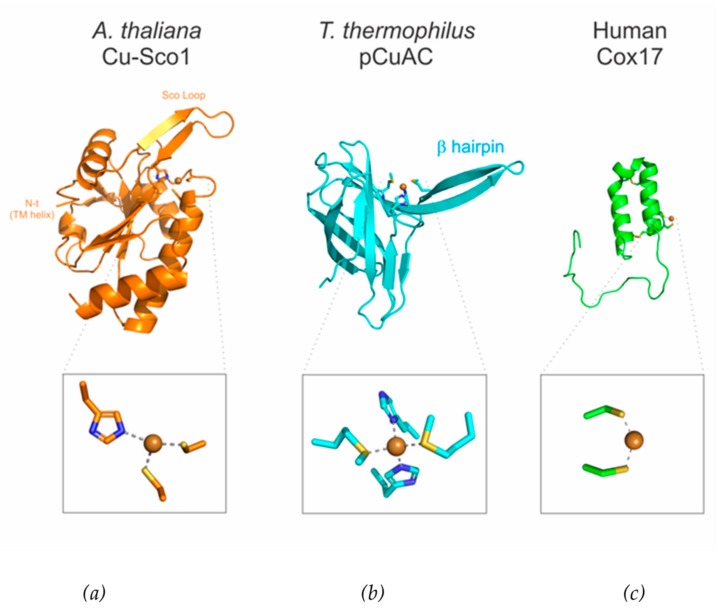
Structures of the relevant proteins in the metallochaperone module: (**a**) X-ray structure of copper-bound *A. thaliana* Sco1 (PDB ID 6N5U) [75] (**b**) NMR structure of copper-bound *T. thermophilus* pCuAC (PDB ID 2K6Z) [74] and (**c**) NMR structure of copper-bound human Cox17 (PDB ID 2RNB) [76]. The insets show the details of the copper sites.

**Table 1 ijms-20-03830-t001:** Assembly factors being part of the metallochaperone module for the Cu_A_ site in different organisms with assigned function.

		Organism	pCuAC Family	Cox17 Family	Sco Family
Prokaryotes	Gram+	*B. subtilis*	--	--	Sco (YpmQ) ^1^
		*S. lividans*	ECuC	--	Sco
	Gram-	*T. thermophilus*	pCuAC	--	Sco
	Alpha- proteobacteria	*P. denitrificans*	pCu1 pCu2	--	ScoA ScoB ^2^
		*B. japonicum*	pCuA ^1^ PCuC ^1^	--	Sco1
Eukaryotes		*S. cerevisiae*	--	Cox17	Sco1 ^1^ Sco2 ^2^
		Human	--	Cox17 ^1^	Sco1 ^1^ Sco2 ^1^
		*A. thaliana*	--	Cox17a ^1^ Cox17b ^1^	Sco1 (Hcc1) ^1^ Sco2 (Hcc2)

^1^ Essential for COX activity. ^2^ Essential under copper-limiting conditions.

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
