# Peer review of "Biochemistry of Copper Site Assembly in Heme-Copper Oxidases: A Theme with Variations"

_ijms, 2019, doi:10.3390/ijms20153830_

Round 1

Reviewer 1 Report

The manuscript depicts de state-of-the-art of Cytochrome c Oxidase assembly, emphasizing the available mechanisms for copper sites assembly. I've found this paper is a chomprehensive, although extensive review of biological mechanisms underlying the assembly of copper sites in COX.

Particularly , I found  the discussion about the role of metallochaperone module essential in unvealing  the sequetial steps for CuA assemby as illustrated in Fig. 3, and the structural details of relevant proteins in the metallochaperone module (Fig. 4) because in this manner both mechanistical/structural  insights become more accesible to a larger audience. Moreover,   I consider the introduction of  short  section  briefly describing CuA assembly process  in mitochodria very useful, coupled with the strategy for the biochemical analysis of mitochondrial metallochaperone module. As a minor comment I would suggest  the authors  to shorten the lenght of the conclusion section and to present the main chalenges regarding the assembly pf COX in different organisms in a more focussed manner.

Author Response

We thank very much the reviewer for the positive comments on our work. We have shortened the conclusions section, and reorganized the main challenges with the aim of making them more clear and easy to read. Also, We have taken care of some grammar and typos throughout the manuscript. 

Reviewer 2 Report

This work aim is to review ultrasensitive new Biochemistry of copper site assembly in heme-copper 2 oxidases: A theme with variations
This review is essential for aerobic respiration and general educational in the field.
They summarize the more common and applicable assembly of COX, with special emphasis in the assembly of copper sites.
The way that the manuscript present is acceptable to me, just an appropriate conclusion which is justified will be useful.
The subject matter is appropriate for the Journal
The quality of the presentation as a review is adequate
There is no apparent lack of clarity
The abstract is informative
There is no material which need to be omitted
There is no apparent errors of fact or logic
Some other appropriate references could be added:
Synthetic Heme/Copper Assemblies: Toward an Understanding of Cytochrome c Oxidase Interactions with Dioxygen and Nitrogen Oxides
Acc Chem Res. 2015 August 18; 48(8): 2462–2474

Author Response

We thank this reviewer for his/her positive comments.

The way that the manuscript present is acceptable to me, just an appropriate conclusion which is justified will be useful.

We thank the reviewer for this suggestion. We have added a paragraph in this regard.

Some other appropriate references could be added:
Synthetic Heme/Copper Assemblies: Toward an Understanding of Cytochrome c Oxidase Interactions with Dioxygen and Nitrogen Oxides
Acc Chem Res. 2015 August 18; 48(8): 2462–2474

We thank he reviewer for the suggestion, the reference has been added.